# Encoding multistate charge order and chirality in endotaxial heterostructures

Samra Husremović [ORCID][1], Berit H. Goodge [ORCID][1,2], Matthew P. Erodici[1], Katherine Inzani [ORCID][3], Alberto Mier[1], Stephanie M. Ribet[4,5,6], Karen C. Bustillo [ORCID][4], Takashi Taniguchi [ORCID][7], Kenji Watanabe [ORCID][8], Colin Ophus [ORCID][4], Sinéad M. Griffin [ORCID][9,10] & D. Kwabena Bediako [ORCID][1,11] ✉

High-density phase change memory (PCM) storage is proposed for materials with multiple intermediate resistance states, which have been observed in $1T$-$TaS_2$ due to charge density wave (CDW) phase transitions. However, the metastability responsible for this behavior makes the presence of multistate switching unpredictable in $TaS_2$ devices. Here, we demonstrate the fabrication of nanothick verti-lateral $H$-$TaS_2$/$1T$-$TaS_2$ heterostructures in which the number of endotaxial metallic $H$-$TaS_2$ monolayers dictates the number of resistance transitions in $1T$-$TaS_2$ lamellae near room temperature. Further, we also observe optically active heterochirality in the CDW superlattice structure, which is modulated in concert with the resistivity steps, and we show how strain engineering can be used to nucleate these polytype conversions. This work positions the principle of endotaxial heterostructures as a promising conceptual framework for reliable, non-volatile, and multi-level switching of structure, chirality, and resistance.

Charge density wave (CDW) materials host correlated electronic states typified by periodic lattice distortions and static modulations of conduction electrons[1]. Non-volatile memory and computing devices based on the principle of phase change memory (PCM)[2,3] may leverage the intrinsic resistivity changes associated with CDW phase transitions[4–9]. $1T$-$TaS_2$, a van der Waals (vdW) layered solid, is a prototypical CDW material in which the atomic lattice distorts in-plane to form 13-atom star-shaped clusters[10,11]. The tiling of these clusters and the extent of commensuration with the underlying atomic lattice define the CDW phases and the electronic properties of $1T$-$TaS_2$[10–15]. Notwithstanding the in-plane nature of this CDW lattice distortion, interlayer coupling plays a key role in stabilizing intralayer charge order in $1T$-$TaS_2$.

Accordingly, together with flake thickness[16–18] and doping levels[16,19], vertical heterostructuring is a powerful route for engineering CDW transitions[20–25]. For example, whereas in pristine, bulk $1T$-$TaS_2$ the commensurate (C) CDW phase only forms below about 180 K[11,12] (and is only observed at much lower temperatures in exfoliated thin flakes[15]), electronically isolating monolayer $1T$-$TaS_2$ with thicker metallic slabs of $H$-$TaS_2$ has been shown to stabilize the C-CDW state in monolayer $1T$-$TaS_2$ at room temperature[23,24]. To this end, the recent synthesis of endotaxial $TaS_2$ offers new mechanisms for accessing modular CDW systems[23].

In this work, we demonstrate an approach converse to preceding literature—employing moderate thermal annealing to interdisperse

[1]Department of Chemistry, University of California, Berkeley, CA 94720, USA. [2]Max-Planck-Institute for Chemical Physics of Solids, Nöthnitzer Str. 40, 01187 Dresden, Germany. [3]School of Chemistry, University of Nottingham, University Park, Nottingham NG7 2RD, UK. [4]National Center for Electron Microscopy, Molecular Foundry, Lawrence Berkeley National Laboratory, Berkeley, CA, USA. [5]Department of Materials Science and Engineering, Northwestern University, Evanston, IL 60208, USA. [6]International Institute of Nanotechnology, Northwestern University, Evanston, IL 60208, USA. [7]Research Center for Functional Materials, National Institute for Materials Science, Tsukuba 305-0044, Japan. [8]International Center for Materials Nanoarchitectonics, National Institute for Materials Science, Tsukuba 305-0044, Japan. [9]Materials Sciences Division, Lawrence Berkeley National Laboratory, Berkeley, CA 94720, USA. [10]The Molecular Foundry, Lawrence Berkeley National Laboratory, Berkeley, CA 94720, USA. [11]Chemical Sciences Division, Lawrence Berkeley National Laboratory, Berkeley, CA 94720, USA. ✉e-mail: bediako@berkeley.edu

monolayer $H$-TaS$_2$ between few-layer $1T$-TaS$_2$ lamellae. In the resulting verti-lateral $1T$-TaS$_2$/$H$-TaS$_2$ heterostructures, decoupled $1T$-TaS$_2$ fragments undergo independent transitions from the disordered incommensurate (IC) to ordered commensurate CDW phase above room temperature. These transitions are hallmarked by synchronous stepwise switching of chirality and resistance with high predictability; the number of steps is encoded by the quantity and arrangement of $H$-TaS$_2$ layers. Thus, the developed materials represent a distinctive framework for deterministic engineering of multistate resistance and chirality changes in $1T$-TaS$_2$. Additionally, we find that the nucleation of $H$-TaS$_2$ polytype initiates at macrosocopic flake defects, a mechanistic insight we harness to showcase the potential of strain engineering for the rational design of verti-lateral TaS$_2$ heterostructures. Moreover, modulating the $H$-TaS$_2$ content tunes the proportion of heterochiral CDW superlattices, resulting in a range of optically detectable net

chiralities. Therefore, our work provides an adaptable roadmap for design of versatile optoelectronic phase change materials with reliable, multilevel and multifunctional switching.

## Results

### Commensuration in verti-lateral TaS$_2$ heterostructures

Exfoliated $1T$-TaS$_2$ crystals were annealed in high vacuum at 350 °C for 30 mins and then rapidly cooled to room temperature (see "Methods" section), engendering changes in optical contrast, indicative of structural transformations (Fig. 1a). This thermally induced phase evolution was characterized by selected area electron diffraction (SAED) of TaS$_2$ flakes before and after annealing. Before annealing, SAED patterns of $1T$-TaS$_2$ flakes are consistent with the expected nearly commensurate (NC)-CDW structure (Fig. 1b)[11]. In contrast, SAED data from annealed samples (Fig. 1c) reveal the

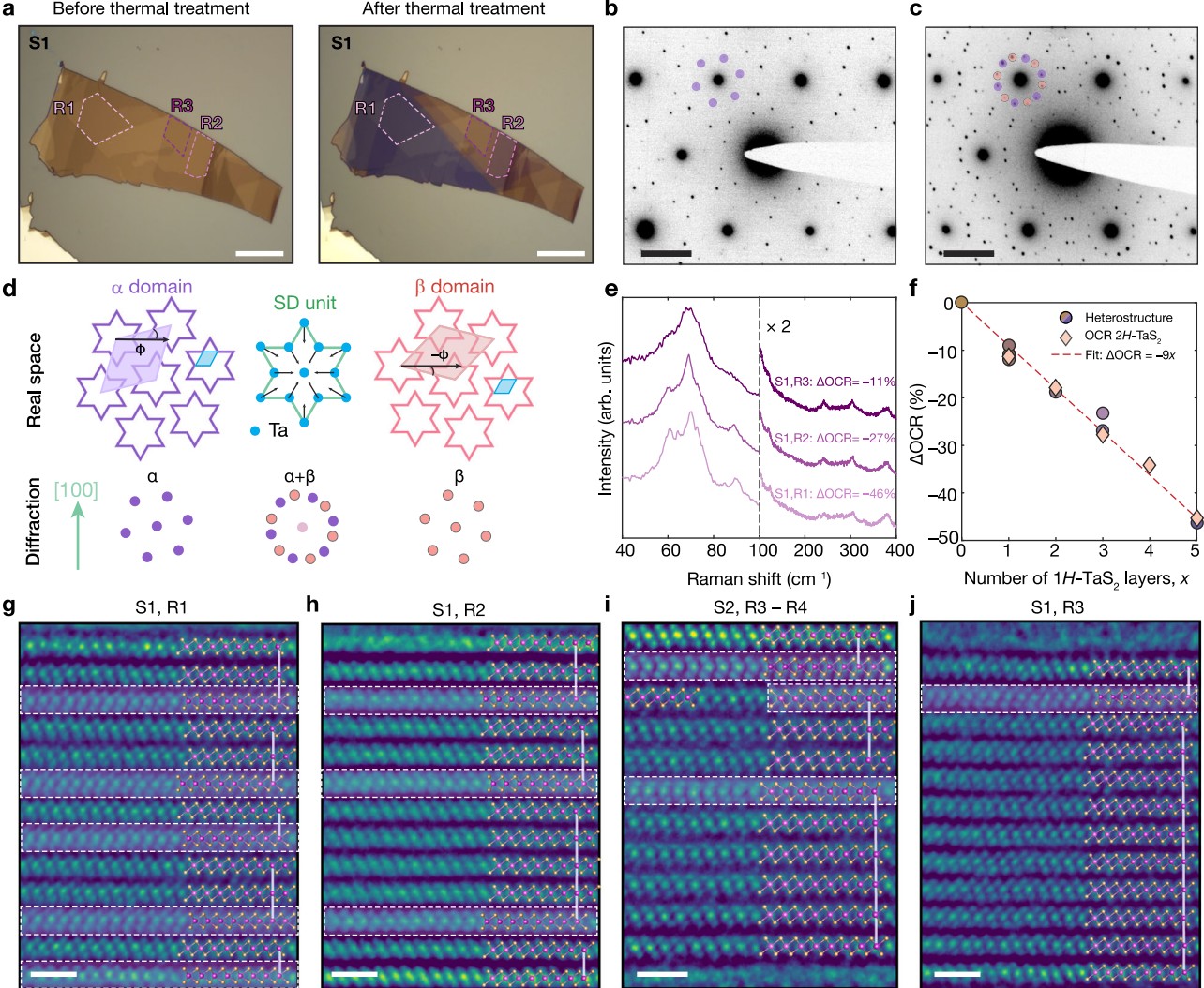

**Fig. 1 | Polytype and charge density wave (CDW) transformations in TaS$_2$ flakes.** **a** Optical micrographs of a $1T$-TaS$_2$ flake, labeled S1, before and after thermal annealing on 90 nm SiO$_2$/Si substrate. Regions of interest are marked. Scale bars: 15 μm. **b, c** Room temperature selected area electron diffraction (SAED) patterns of a representative $1T$-TaS$_2$ flake before (**b**) and after (**c**) thermal annealing. A representative set of first-order superlattice peaks is marked for each sample. Scale bars: 2 nm$^{-1}$. **d** Schematic of real space $\alpha$ and $\beta$ CDW domains. Unit cells of $\alpha$, $\beta$, and $1T$-TaS$_2$ are shaded in violet, pink and blue, respectively. Angle $\phi$ represents the rotational misalignment between the unit cell of $1T$-TaS$_2$ and the unit cells of the CDW superlattices. Schematic diffraction patterns for $\alpha$, $\beta$, and $\alpha + \beta$ patterns are shown at the bottom. **e** Linearly polarized Raman spectra in selected regions of S1

after thermal annealing. Vertical dashed line represents the energy cutoff after which the Raman intensity was scaled by a factor of 2. **f** Relationship between the red-channel optical contrast change upon annealing ($\Delta$OCR) and the number of $H$-TaS$_2$ layers in distinct regions of S1 and S2 (a $1T$-TaS$_2$ sample) after annealing. This data is overlaid with the thickness dependence of OCR for $2H$-TaS$_2$ on 90 nm SiO$_2$/Si substrate (ref. 31). **g–j** Atomic-resolution differential-phase-contrast scanning transmission electron microscopy (DPC-STEM) images along the [10$\bar{1}$0] zone axis of: region R1 of S1 (**g**), area R2 of S1 (**h**), regions R3--R4 of S2 (**i**), and region R3 of S1 (**j**) overlaid with structures of $1T$-TaS$_2$ (ref. 63) and $H$-TaS$_2$ (ref. 64). $H$-TaS$_2$ layers are highlighted and Ta ions aligned along the $c$-axis are connected with pink lines. Scale bars in **g–j** are 1 nm.

presence of two $\sqrt{13} \times \sqrt{13}$ heterochiral superlattices. The CDW enantiomorphs, $\alpha$ (L) and $\beta$ (R), possess supercells that are rotated $\pm 13.9$ degrees relative to the unit cell of $1T$-TaS$_2$ (Fig. 1d)[11,26,27]. Additionally, the SAED data of annealed $1T$-TaS$_2$ flakes are consistent with a C-CDW phase at room temperature[23,24]. These ensemble changes in extent of CDW commensuration in heat-treated flakes were also evinced by linearly polarized Raman spectroscopy (Fig. 1e and Supplementary Figs. 15 and 16). Raman spectra obtained in visually distinct regions of annealed flake S1 (Fig. 1e) show a strong correlation between the red-channel optical contrast change upon annealing ($\Delta$OCR) and the sharpness of low-frequency (40–100 cm$^{-1}$) Raman modes related to CDWs[28–30]. The CDW spectral peaks become increasingly well-defined from regions 3 to 1 (R3–R1), a behavior that is tightly correlated with progressive $\Delta$OCR. Furthermore, as $\Delta$OCR becomes more negative, new Raman spectral features emerge at 64 cm$^{-1}$, 89 cm$^{-1}$, 123 cm$^{-1}$, and 230 cm$^{-1}$. The manifestation of new peaks and general sharpening of Raman features is consistent with Brillouin zone folding of $1T$-TaS$_2$ as it undergoes the transition from the nearly commensurate CDW (NC-CDW) to C-CDW[28–30].

To unveil the relationship between $\Delta$OCR, CDW commensuration, and atomic lattice structure of these annealed $1T$-TaS$_2$ crystals, atomic resolution differential-phase-contrast scanning transmission electron microscopy (DPC-STEM) imaging and analysis (Fig. 1f–j) was performed on cross-sectional samples made from optically distinct regions of flakes S1 (Fig. 1a) and S2 (Supplementary Fig. 15b). Representative DPC-STEM micrographs, depicted in Fig. 1g–j, reveal that thermal annealing induces a partial transition from the $1T$ (octahedrally coordinated Ta) to the $H$ (trigonal prismatic Ta) structure. The $H$ polytype forms within the $1T$-TaS$_2$ matrix (i.e., endotaxially), overwhelmingly as monolayers, separating generally thicker fragments of $1T$-TaS$_2$. At boundaries between regions with dissimilar $\Delta$OCR, the number of $H$-TaS$_2$ layers varies across a single $1T$-TaS$_2$ flake, constructing lateral heterostructures with atomically sharp interfaces (Fig. 1i). Furthermore, these DPC-STEM data show how $1T$-TaS$_2$ slabs separated by a $H$-TaS$_2$ layer slip relative to each other, resulting in the misalignment of the Ta centers (vertical lines in Fig. 1g–j) in mixed polymorph heterostructures.

We find that optical contrast measurements are a powerful and convenient tool for identifying polymorph composition, owing to the linear relationship between $\Delta$OCR upon annealing and the number of $H$-TaS$_2$ layers, determined from DPC-STEM (Fig. 1f). This relationship mirrors, and stems from, the linear scaling between layer count and OCR for freestanding few-layer $H$-TaS$_2$[31] (see Supplementary Note 2 for details). Therefore, crystal sections with a more negative $\Delta$OCR (i.e., appearing increasingly blue) contain a greater number of layers of the $H$ polymorph. Taken together with the increased sharpness and number of CDW Raman modes (Fig. 1e), these data establish that the formation of monolayer $H$ structures leads to increased order of $1T$-TaS$_2$ C-CDW domains, consistent with prior work with thicker $H$-TaS$_2$ slabs[23,24] and bulk mixed polytype TaS$_2$ phases[12,26,32,33].

The ensemble room-temperature ordering of CDW domains in TaS$_2$ heterostructures with different polytype compositions was further probed using SAED. Our findings support that samples comprising $\geq 20\%$ of $H$-TaS$_2$ manifest an ordered C-CDW phase, characterized by sharp CDW reflections (Supplementary Fig. 4b–d). In contrast, the CDW reflections observed in samples with $< 20\%$ $H$-TaS$_2$ appear less well-defined and exhibit peak splitting and angular blurring (Supplementary Fig. 4a). Accordingly, we infer that crystals containing less than 20% $H$-TaS$_2$ content host a disordered C-CDW phase with likely coexistence of some NC-CDW domains. Notably, the chirality of CDW domains (Fig. 1d) could be exploited for next-generation switching devices, making the nano-scale structural understanding of C-CDWs in TaS$_2$ heterostructures integral for their potential applications.

## Diverse chirality in polytype heterostructures

The heterochirality of CDW superlattices in $H$-TaS$_2$/$1T$-TaS$_2$ at room temperature was mapped with nano-scale resolution using four-dimensional scanning transmission electron microscopy (4D-STEM), establishing that the $\alpha$ and $\beta$ enantiomorphs are stacked along the $c$-axis to form vertical CDW superstructures. In 4D-STEM, a converged electron probe is scanned across a sample in a 2D array, while recording 2D diffraction data at each probe position (Fig. 2a)[34]. We obtained 4D-STEM datasets of a 20-layer flake, S3 (Fig. 2b), parallel to the $c$-axis with ~5.5 nm spatial resolution in regions R2–R5. These regions exhibit progressively sharper Raman spectral features (Supplementary Figure 16) and larger $H/T$ ratios from measurements of $\Delta$OCR. For R2–R5, Bragg reflections associated with both commensurate $\alpha$ and $\beta$ enantiomorphs are present in nanodiffraction patterns (Fig. 2c, d and Supplementary Figure 7). Thus, the CDW enantiomorphs are coexistent in a ~5.5 nm area, revealing their formation in the out-of-plane ($\|c$-axis) direction. We note that CDW superstructures can be vertically stacked in two configurations across the $H$-TaS$_2$ interface: heterochiral ($\alpha$–$\beta$) (Fig. 2e) and homochiral ($\alpha$–$\alpha$) (Fig. 2f), each hosting inequivalent CDW cluster interlayer stackings and charge distributions (Supplementary Figure 19)[27]. The interlayer arrangement of CDW clusters in polytype heterostructures exhibits notable distinctions compared to both $1T$-TaS$_2$ flakes and homointerfaces. In pristine $1T$-TaS$_2$, CDW clusters can be perfectly eclipsed because their building blocks—Ta ions—are directly aligned in the out-of-plane direction[35,36]. In contrast, in polytype heterostructures, Ta ions in $1T$-TaS$_2$ slabs across $H$-TaS$_2$ interfaces must be laterally offset (Fig. 1g–j). Thus, CDW clusters in neighboring $1T$-TaS$_2$ slabs must assume a staggered arrangement, resulting in distinct CDW superlattice patterns (2e–f, Supplementary Figure 19).

Next, integrated intensities of $\alpha$ and $\beta$ diffraction spots were evaluated at each probe position (see "Methods" section for details)[37] to reconstruct dark-field images associated with the difference in superlattice ratios, defined as: $(\beta - \alpha)/(\beta + \alpha)$ (Fig. 2g–j). The resultant enantiomorphic ratio maps reveal a minor extent of in-plane variation within each heterostructure (see histograms in Fig. 2 g–j), potentially arising from domain pinning defects. Nevertheless, 4D-STEM data and high-resolution TEM analysis (Supplementary Figure 5) consistently point to coexisting, out-of-plane $\alpha$ and $\beta$ superstructures regardless of the $H/T$ composition. However, the polymorph composition appears to influence the $\alpha/\beta$ proportion. For example, a less than 45% mean difference in ratio of diffraction pattern intensity from enantiomorphic phases was measured for R4 and R5 at room temperature (Fig. 2i, j). Conversely, a larger mean enantiomorphic disproportion exceeding 65% was obtained for R2 and R3 (Fig. 2g, h). This can be understood by considering that changing the polymorph composition alters the size and number of $1T$-TaS$_2$ fragments hosting the two heterochiral CDWs, thereby engendering changes in the overall chirality. A high enantiomorphic disproportion is therefore the most likely for less transformed samples with larger $1T$-TaS$_2$ fragments, which would then dominate the overall chirality.

The chirality of TaS$_2$ heterostructures can also be assessed optically, as enantiomorphic disproportion engenders Raman optical activity (ROA): a distinct Raman response to right- and left-circularly polarized light (see Supplementary Note 9 for details)[36,38]. For $1T$-TaS$_2$ displaying ROA, the integrated area ratio of $E_g$(I) and $E_g$(II) modes differs in the $\sigma^+\sigma^-$ and $\sigma^-\sigma^+$ Raman polarization configurations (Fig. 3a), where $\sigma^i\sigma^s$ ($i, s = \pm$) are phonon helicities of the incident and scattered light[38]. Note, a stronger ROA signals a higher enantiomorphic disproportion and a larger overall chirality. Representative chirality-dependent optical measurements of sample S4 (Supplementary Figure 18c) are shown in Fig. 3. We find that ROA decreases from region R1 ($x/n = 3/20$) to R3 ($x/n = 5/20$), signaling decreasing overall chirality for an increasing $H$-TaS$_2$ layer count (Fig. 3b–d and Supplementary Fig. 14). These results are consistent with our 4D-STEM findings; highly

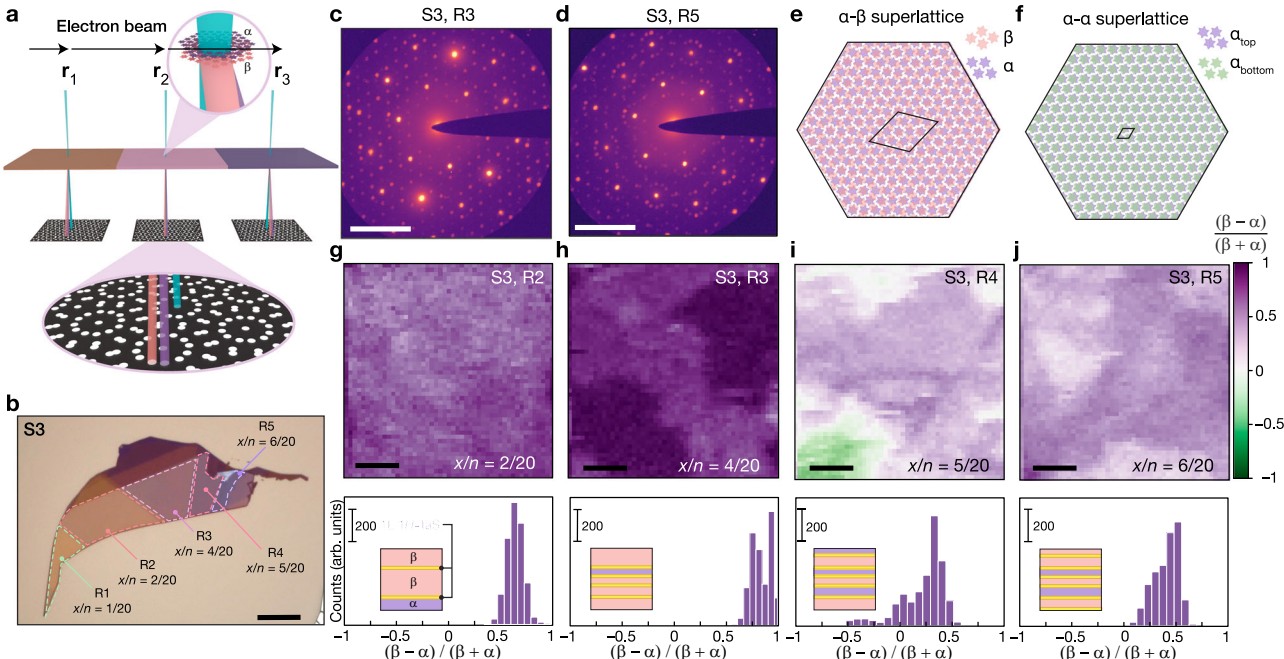

**Fig. 2 | Mapping heterochiral CDW domains with four-dimensional scanning transmission electron microscopy (4D-STEM). a** Schematic illustrating 4D-STEM of annealed 1*T*-TaS₂ samples. Three compositionally distinct flake regions are labeled as $r_1$, $r_2$ and $r_3$. The two heterochiral superlattices are marked as $\alpha$ and $\beta$. **b** Optical micrograph of a 20-layer annealed 1*T*-TaS₂ sample S3. Regions of interest and their *H*-TaS₂ proportion ($x/n$, where $x$ = number of *H*-TaS₂ layers and $n$ = total number of layers), calculated from optical contrast measurements, are labeled. Scale bar: 10 μm. **c, d** The maximal diffraction patterns, displayed on a logarithmic scale, for S3--R3 (**c**) and S3--R5 (**d**). Scale bars: 5 nm⁻¹. **e, f** Illustration of CDW

superstructures for $\alpha,\beta$ (**e**) and $\alpha,\alpha$ (**f**). Unit cells of the CDW superstructures are outlined in black. **g--j** $(\beta-\alpha) / (\beta+\alpha)$ virtual dark-field images and their histograms for R2--R5 regions of S3. Insets show the proposed sample composition and plausible chirality stacking based on the *H*-TaS₂ content and 4D-STEM analysis of each region. We note that only the net chirality can be determined, and the exact stacking sequence of the chirality shown here is one of several possibilities as the precise sequence cannot be obtained from plan-view 4D-STEM. All 4D-STEM data was acquired at room temperature. Scale bars in **g--j**: 50 nm.

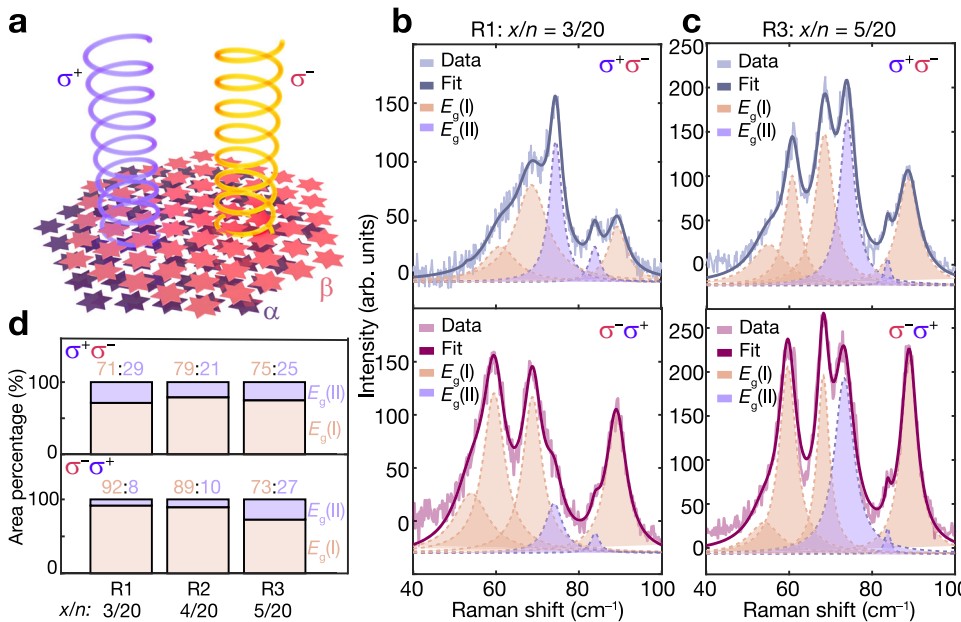

**Fig. 3 | Optical detection of chirality in TaS₂ heterostructures. a** Schematic of a polarization-dependent Raman measurement. **b, c** Raman spectra in the circular contrarotating polarization configurations ($\sigma^+\sigma^-$ and $\sigma^-\sigma^+$) obtained for region R1 (**b**) and region R3 (**c**) of a 20-layer sample S4. Lorentzian peak fits and the cumulative

fits are displayed. **d** Stacked bar charts displaying the normalized area percentage of $E_g(I)$ and $E_g(II)$ modes measured in the two contrarotating Raman polarization configurations (top: $\sigma^+\sigma^-$, bottom: $\sigma^-\sigma^+$) for R1–R3 of S4. For **b–d**, $x/n$ denotes the *H*-TaS₂ proportion. Data was obtained at room temperature.

transformed regions, on average, exhibit a weaker overall chirality compared to medium transformed regions (Fig. 2g–j). Notably, our verti-lateral heterostructures exhibit a broad spectrum of possible overall chiralities, distinguishing them from $1T$-TaS$_2$ flakes/homo-interfaces and heavily transformed $H$-TaS$_2$/$1T$-TaS$_2$ heterostructures, which can only manifest a single enantiomorphic state[23,36]. Specifically, the former are homochiral, comprising fully of $\alpha$ or $\beta$[36], while the latter have been shown to be achiral, hosting an equal proportion of $\alpha$ and $\beta$[23]. Accordingly, our verti-lateral heterostructures may enable chiral opto-electronic memory schemes through their wide array of chiral states that can generate strong optical responses at room temperature.

## Multistate resistance and chirality switching

Electronic properties of these heterochiral endotaxial polytype heterostructures were probed using variable-temperature transport measurements. In these studies, we monitored the temperature-dependent longitudinal resistance ($R_{xx}$) of mesoscopic devices fabricated from S1 (Fig. 4a) and S4 (Supplementary Fig. 18c). Measurements from compositionally distinct regions (Fig. 4b) were used to establish the relationship between polytype composition and the IC-to-C CDW phase transition behavior. For all measured regions, transport below 300 K is dominated by the metallic $H$-TaS$_2$ layers (decreasing resistance with decreasing temperature), while transport above room temperature traces CDW transitions of $1T$-TaS$_2$ lamellae (Fig. 4c,d). These transport data were complemented with temperature-dependent 4D-STEM of representative heterostructures to provide structural insight (Fig. 4e–g). Above room temperature, $1T$-TaS$_2$

transforms from the IC (more conductive) to the C (more insulating) CDW phase with a hysteresis between cooling and warming profiles (Fig. 4c–e)[11]. As a general observation, increasing $H$-TaS$_2$ content leads to narrowing of the thermal hysteresis (Fig. 4c), which indicates stabilization of the ordered C-CDW state. These observations lie in agreement with our confocal Raman measurements; consistently sharper CDW Raman features are observed for samples with a higher $H$-TaS$_2$ content (Fig. 1e and Supplementary Figs. 15 and 16).

Interestingly, the formation of $H$-TaS$_2$ layers dictates the stepwise evolution of resistance and chirality with temperature in these endotaxial polytype heterostructures. Upon cooling, we observe a series of stepped resistance increases (Fig. 4d), with the step count matching the number of $1T$-TaS$_2$ slabs determined from DPC-STEM images of device cross-sections. Thus, the $H$-TaS$_2$-separated $1T$-TaS$_2$ lamellae behave as isolated crystals with distinct IC-C CDW transitions, likely due to the electronic decoupling imposed by the metallic ($H$-TaS$_2$) spacers[23]. For this reason, the number of $H$ spacer layers deterministically encodes the number of resistance steps, and the profile (magnitude of resistivity change) of each step is governed by the various thicknesses of the $1T$-TaS$_2$ segments; thicker $1T$-TaS$_2$ slabs exhibit sharper CDW transitions, as observed in freestanding $1T$-TaS$_2$ crystals[17,18]. We note that resistance traces upon warming are noticeably broadened relative to the respective cooling sweeps. This may be understood by considering that strength of defect pinning is contingent on the CDW phase[15,17,28,39]. Defects may exert stronger pinning effects on the CDW domains in the localized C-CDW state compared to the "melted" IC-CDW phase, leading to less well-defined transitions upon warming. Nevertheless, the stepwise resistance transitions are

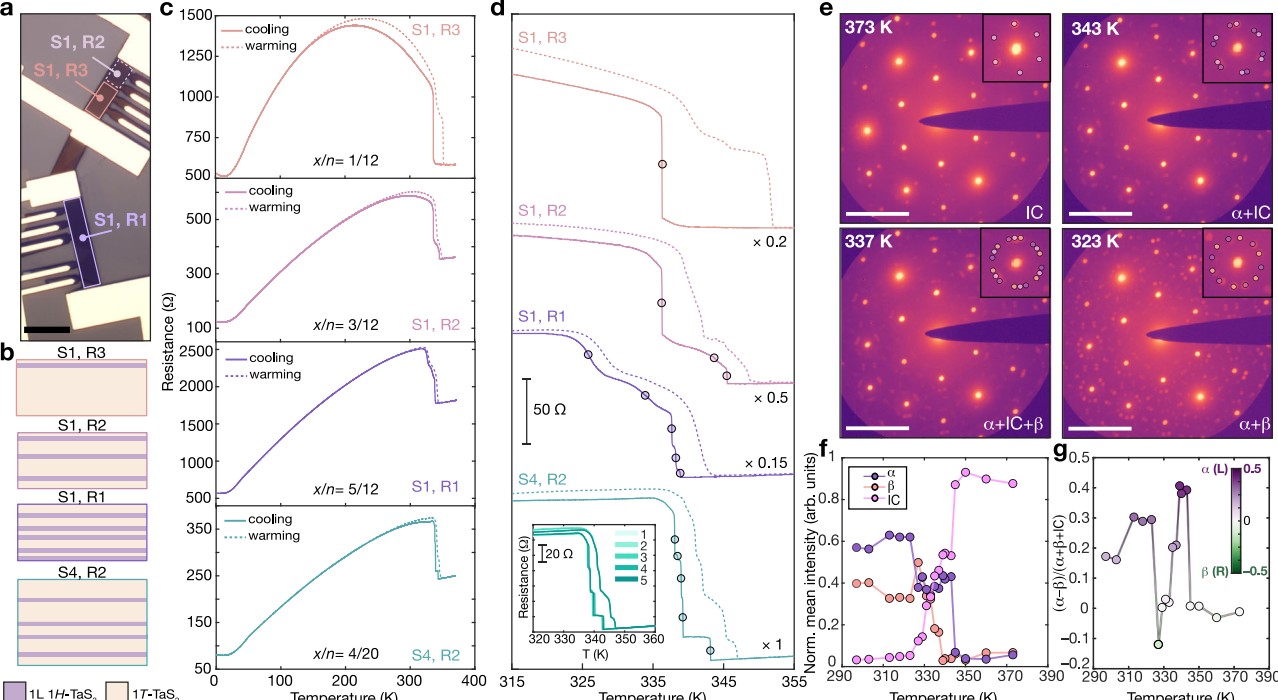

**Fig. 4 | Multistate resistance and chirality switching in TaS$_2$ heterostructures. a** Optical micrograph of a mesoscopic device fabricated from annealed flake S1. Scale bar: 10 μm. **b** Diagrams of sample composition for S1 and S4. For S1, diagrams are derived from atomic resolution DPC-STEM data, while for S4 the structure is proposed based on the ΔOCR-derived $H$-TaS$_2$ content. Note, for S4, the $H$-TaS$_2$ layers are randomly placed in the model. **c** Temperature-dependent resistance of S1 in regions R1–R3 and S4 in region R2. The marked $x/n$ denotes the $H$-TaS$_2$ proportion. **d** High-temperature section of **c**. Curves were vertically shifted for clarity. Inflection points in the cooling curve are marked by a circle. Scaling factors of the resistance curves are indicated on the right. Inset shows the resistance modulation

of S4--R2 in five thermal cycles. Temperature ramp rate in **c**, **d** was 1 K/min. **e** Representative temperature-dependent 4D-STEM diffraction patterns, displayed on a logarithmic scale, for a 20-layer heterostructure with five $H$-TaS$_2$ lamella. Insets display a zoomed-in view of a primary Bragg spot with labeled CDW superstructure peaks. Scale bars: 5 nm⁻¹. **f** Normalized mean intensity of $\alpha$, $\beta$ and incommensurate (IC) diffraction peaks in 4D-STEM datasets obtained at different temperatures for the heterostructure in **e**. **g** Temperature-dependent $(\alpha - \beta)/(\alpha + \beta + IC)$ calculated from **f**. For **e**–**g**, heating was performed in situ and the data were acquired upon cooling from 373 K at 1 K/min.

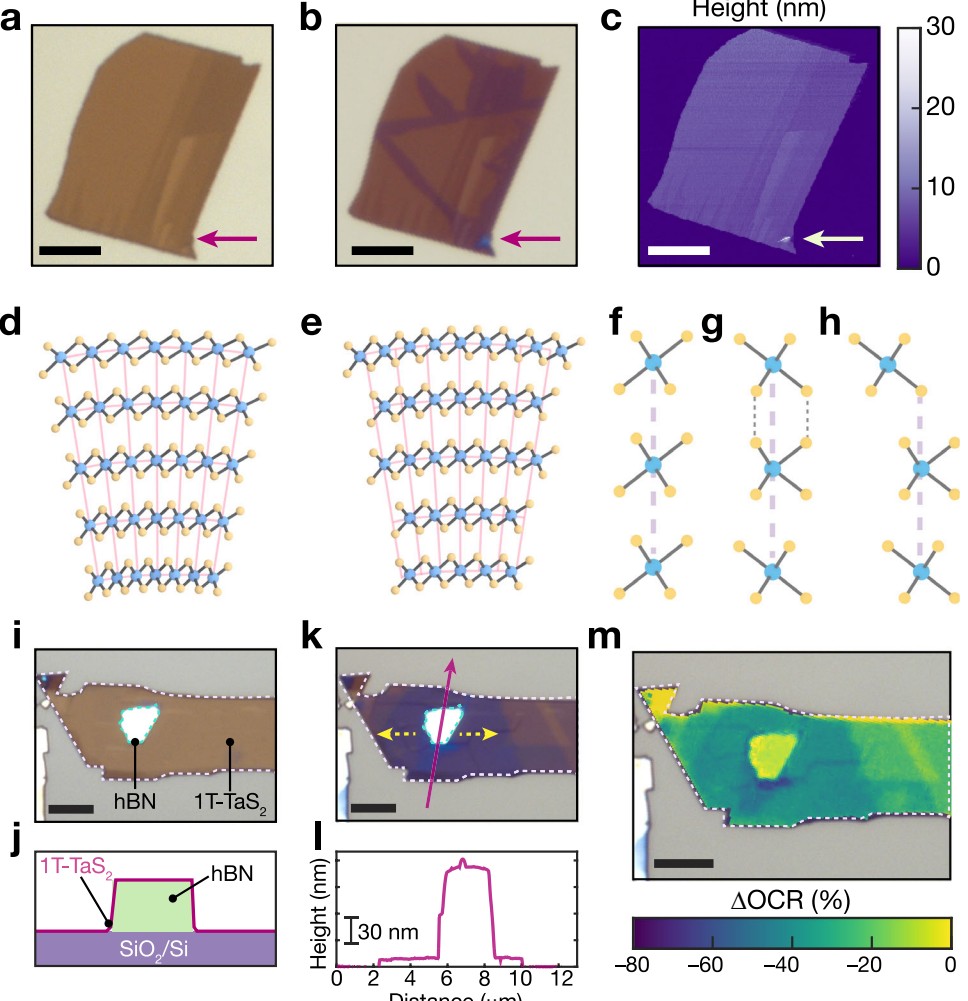

**Fig. 5 | Nucleation of endotaxial polytype transformation. a, b** Optical image of a 1*T*-TaS$_2$ flake before (**a**) and after (**b**) thermal annealing. A flake fold is marked with a magenta arrow. **c** Atomic Force Microscopy (AFM) map of **b**. **d, e** Models of 1*T*-TaS$_2$ bending at a fold or wrinkle that are accommodated by in-plane strain (**d**) or interlayer shear and slip **e**. Models in **d** and **e** are adapted from ref. 42. **f–h** Stacking configurations of three TaS$_2$ structures: interface of three 1*T*-TaS$_2$ layers (**f**), 1*T*-TaS$_2$/1*H*-TaS$_2$/1*T*-TaS$_2$ interface without layer sliding (**g**) and 1*T*-TaS$_2$/1*H*-TaS$_2$/1*T*-TaS$_2$ interface with shifting of the topmost 1*T*-TaS$_2$ layer (**h**). Violet dashed vertical lines in **f–h** indicate the alignment of Ta ions between the three TaS$_2$ layers. Black dashed lines in (**g**) represent direct vertical overlap of sulfur ions between the topmost and middle layers. **i, j** Optical micrograph (**i**) and illustration (**j**) of an hBN/1*T*-TaS$_2$ heterostructure before annealing. **k** Optical micrograph of flake in **i** after annealing. Yellow arrows indicate the direction of propagation of the blue *H*-TaS$_2$ domains. **l** AFM height profile of flake in **i**, **k** in the region marked with a magenta arrow. **m** Map of ΔOCR after annealing overlaid on the optical micrograph (**k**). Flake outlines are added in **i**, **k**, and **m**. All scale bars are 5 μm.

highly reproducible in subsequent cooling cycles (Fig. 4d and Supplementary Figs. 17 and 18d). Moreover, in addition to the resistance steps upon cooling, temperature-dependent 4D-STEM reveals stepwise appearance of the $\alpha$ (L) and $\beta$ (R) C-CDW enantiomorphs in concert with vanishing of the achiral IC phase (Fig. 4e–g). Thus, the IC-C CDW transition in our endotaxial heterostructures is marked both by simultaneous evolution of resistance and overall chirality (Fig. 4g), defined by the proportion of chiral superlattices: $(\alpha-\beta)/(\alpha+\beta+IC)$. This synchronous switching sets the stage for optoelectronic devices combining charge and chirality degrees of freedom.

## Polytype nucleation and designer heterostructures

Lastly, having established the structure and multistate electronic/chiral switching in TaS$_2$ endotaxial heterostructures, we turn to considering the mechanism of nucleation of these polytype transformations, finding them to be facilitated by wrinkles and folds. We observed that polymorph transitions in 1*T*-TaS$_2$ flakes, evidenced by changes in ΔOCR and sharpness of Raman spectral features associated with CDW modes, generally emanate from wrinkles, tears, and folds in flakes

(Fig. 5a–c and Supplementary Fig. 20). These microscale structural defects inevitably introduce differential stress, which can be accommodated by strain (Fig. 5d)[40–42], and this strain in turn can alter the energetic barrier between polytypes[43–46]. Accordingly, one explanation for the emanation of *H*-TaS$_2$ at these features, is a strain-induced decreased energetic barrier for the *H−T* transformation, facilitating in nucleation of polytypic domains near stress points. In addition, differential stress in layered materials can also be accommodated by shear and slip between layers (Fig. 5e)[42]. This leads to the formation of extended shear dislocations[42], which as we observed in Fig. 1g–j, appear to be a prerequisite for the formation of 1*T*-TaS$_2$/1*H*-TaS$_2$/1*T*-TaS$_2$ interfaces from 1*T*-TaS$_2$[32]. Specifically, in native 1*T*-TaS$_2$, the Ta ions are directly aligned in the out-of-plane direction (Fig. 5f). However, upon transformation of one layer to *H*-TaS$_2$, crystallographic stacking with direct S−S overlap would be encountered (Fig. 5g). We find this configuration to be, on average, 8 meV per Å$^2$ higher in energy according to density functional theory calculations (see "Methods" section for calculation details). To eliminate this unfavorable interlayer interaction, one of the 1*T*-TaS$_2$ layers can slip across the trigonal

prismatic interface (Fig. 5h), precisely as observed in Fig. 1g–j. Note that interlayer slips are readily present at flake folds, wrinkles and tears in 2D materials. Thus, polytype domains may form more readily in those defect regions of 1T-TaS$_2$. We surmise that polytype transitions nucleate at macroscopic defect points due to the formation of extended dislocations and/or decreasing of the 1T-H energy barrier due to strain.

We build upon these nanoscale insights to demonstrate that mechanical/strain engineering of 1T-TaS$_2$ may be used for rational design of vertical/lateral TaS$_2$ heterostructures. To this end, we stacked a 14-layer 1T-TaS$_2$ crystal onto a ~112 nm-thick hexagonal boron nitride (hBN) flake to deliberately impart local stress onto the region of the 1T-TaS$_2$ flake in the vicinity of hBN (Fig. 5i,j). Indeed, after annealing, the H-TaS$_2$ polytype formation, evidenced by ΔOCR, radiates away from the hBN/TaS$_2$ interface (Fig. 5k–m). It is important to note that coincident vertical/lateral heterostrucures are only formed upon annealing at moderate temperatures for short time periods, as extended or repeated heating leads to formation of predominantly homogeneous structures (Supplementary Fig. 21)[23]. Accordingly, intricate, multi-component device architectures could be realized by combining substrate patterning with moderate thermal annealing conditions.

## Discussion

In conclusion, we have demonstrated that highly tunable, verti-lateral polytype heterostrucures of 1T-TaS$_2$ and H-TaS$_2$ can be synthesized by moderate thermal annealing of nano-thick 1T-TaS$_2$ flakes. Stress points in 1T-TaS$_2$ are nucleation sites for the H-TaS$_2$ domains, resulting in multi-component flakes with coexisting vertical and lateral heterostructures. The polytype composition of these heterostructures can now be conveniently determined using the optical contrast methodology developed in this work, enhancing the accessibility for characterizing and studying complex TaS$_2$ crystals. Further, we use electron microscopy, Raman spectroscopy, and electronic transport studies to show that altering the H/1T ratio opens manifold possibilities for tailoring structural and electronic behavior of mixed polymorph crystals. Interlayer coupling between 1T-TaS$_2$ fragments is interrupted by H-TaS$_2$ layers, and the increasing content of H-TaS$_2$ correlates with greater CDW commensuration, the appearance of optically detectable heterochiral CDW superlattices, and tunable chirality by modulating the $\alpha/\beta$ ratios. Furthermore, decoupled 1T-TaS$_2$ fragments transition from the IC-CDW to the C-CDW state independently at high temperatures and the resulting temperature-driven, multistate phase transition is highly predictable: the number and size of resistance and steps is defined by the count and placement of H-TaS$_2$ layers within the polytype heterostructure. Moreover, the changes in resistance are accompanied by corresponding changes in chirality, resulting in simultaneous modifications of both optical and electrical properties. Given the precedent for fast switching in 1T-TaS$_2$ using electrical[5,7,15,27,47–50] and optical fields[35,51], endotaxial H-TaS$_2$/1T-TaS$_2$ heterostructures offer a rich framework as designer CDW materials that should exhibit multi-level switching between chiral CDW phases. The ability to predictably fabricate and control such well-defined phase change materials using low-energy external stimuli is a highly promising roadmap toward next-generation computing and data storage.

## Methods

### Mechanical exfoliation of 1T-TaS$_2$

The mechanical exfoliation of 1T-TaS$_2$ (HQ Graphene) is done in an Ar glovebox using an adhesive tape (Magic Scotch). The crystals are exfoliated onto 90 nm SiO$_2$/Si wafers, whose oxide layer is formed by dry chlorination followed by annealing in forming gas (Nova Electronic Materials). First, wafers are cut into ~1 × 1 cm pieces and cleaned for 2 mins in an oxygen plasma cleaner. The chips are then heated to 200 °C on the glovebox hotplate while tessellating a sizeable (~3 × 3 mm) 1T-TaS$_2$ crystal with the adhesive tape. Following this, chips are taken off the hotplate, and placed shiny-side-up onto the tape while still warm. The chips are then pressed for 10 mins with finger pressure. Lastly, the tape is swiftly taken off the chips.

### Mechanical exfoliation of hexagonal boron nitride

The mechanical exfoliation of hBN (used as received from T. Taniguchi and K. Watanabe) is performed in atomospheric conditions. First, a 90 nm SiO$_2$/Si wafer (Nova Electronic Materials) is cut into ~1 × 1 cm pieces and cleaned for 90 mins in an ozone cleaner. Immediately before the chip cleaning is complete, 3 hBN crystals (~1.5 mm × 1.5 mm) are tessellated with an adhesive tape (Magic Scotch). Promptly after cleaning the chips, they are placed shiny-side-up onto the tape and pressed for 10 mins with finger pressure. After this, the tape is swiftly taken off the chips.

### Thermal annealing of 1T-TaS$_2$ crystals

The 1T-TaS$_2$ flakes were annealed in high-vacuum (approximately 10$^{-7}$ Torr) by rapidly warming to 120 °C at 60 °C/min with a 5-minute hold. This is followed by heating at 11.5 °C/min to 350 °C and a 30 min hold at 350 °C, before rapidly cooling to room temperature at 13.5 °C/min.

### Determination of layer count for TaS$_2$ flakes

For flakes S1 and S2 that were studied by differential-phase-contrast scanning transmission electron microscopy (DPC-STEM)[52], we counted the number of layers in the atomic resolution data. For all other flakes, the number of layers was identified using optical contrast (OC) measurements, following the relationship between OC and thickness established in ref. 53. The OC data was commonly obtained in conjunction with atomic force microscopy (AFM) to corroborate the OC-derived thickness.

### Preparation of samples for transmission electron microscopy

For c-axis imaging, we prepared our samples within an Ar glovebox using a custom-built transfer stage. This involved using a polymeric stamp composed of a poly(bisphenol A carbonate) (PC) film covering a polydimethylsiloxane (PDMS) square on a glass slide. The PC/PDMS stamp was created by initially preparing a solution of PC in chloroform with a concentration of 5 % w/w. The solution was then dispensed onto a glass slide using a pipette and evenly distributed by placing it between two glass slides, which were promptly separated. Subsequently, the slides were positioned with the PC side facing up on a hotplate set at 120 °C for 5 mins, resulting in the formation of a relatively uniform PC film. This PC film was then precision-cut into small squares, ~2 mm × 2 mm in size, using a razor blade. Additionally, squares of PDMS, measuring approximately 4 mm × 4 mm, were prepared and affixed to glass slides. To complete the stamp, a PC square was centered within the PDMS square. Finally, the stamp was placed with the polymer side facing up on a hotplate set at 120 °C for 2 mins. Next, the PC/PDMS stamp is used to pick up 1T-TaS$_2$ flakes. To this end, 1T-TaS$_2$ flakes of interest were covered in PC for 3 mins at 160 °C, followed by rapid cooling to room temperature. Then, flakes adhered to the PC/PDMS stamp were placed onto a 200 nm silicon nitride holey TEM grid (Norcada) by melting the PC polymer at 160 °C. The TEM grid underwent a 5-minute cleaning process with O$_2$ plasma immediately before stacking to enhance flake adhesion. After stacking, the PC film was dissolved in chloroform for 20 mins under ambient atmosphere, washed in isopropanol and dried with flowing N$_2$.

For imaging along the crystallographic ab-plane, cross-sectional TEM samples were prepared by standard the standard focused ion beam (FIB) lift-out procedure using Thermo Fisher Scientific Helios G4 and Scios 2 FIB-SEM systems. A 200 nm coating of Pt was deposited over the region of interest using an electron beam at 5 kV and 1.6 nA. This was followed by a deposition of 2.5 $\mu$m Pt using a gallium-ion

beam at 8 kV and 0.12 nA. The initial lift-out was performed with a 30 kV Ga beam and 3 nA probe current, followed by a 1 nA current for the lift-out cleaning. Next, the sample was milled with 30 kV and 0.5 nA until reaching ~ 1 μm thickness, followed by 16 kV and 0.23 nA thinning to 0.5 μm. Next, a 5 kV beam, operated between 77–48 pA, was used to thin the sample to electron transparency. Lastly, final polishing was done at 2 kV and 43 pA.

## Nanofabrication of mesoscopic devices from $TaS_2$ heterostructures

All nanofabrication steps were performed in the Marvell Nanofabrication Laboratory. In a typical procedure, electron beam lithography (100 kV Crestec CABL-UH Series Electron Beam Lithography System) was used to define electrical contacts. PMMA (polymethyl methacrylate) e-Beam resist was used for this purpose (950 PMMA A6, Micro-Chem). After lithography, reactive ion etching (RIE) with a mixture of 70 sccm $CHF_3$ and 10 sccm $O_2$ (Semigroup RIE Etcher) was used to remove top $TaS_2$ layers and expose a fresh surface immediately before evaporating Cr/Pt (1 nm/100 nm) (NRC thermal evaporator). After overnight metal lift-off in acetone, electron-beam lithography was used to define a Hall bar-shaped etch mask. Etching of the Hall bar was accomplished by reactive ion etching with 70 sccm $SHF_6$ and 8.75 sccm $O_2$ (Semigroup RIE Etcher). After the $SHF_6/O_2$ treatment, samples were cleaned for 20 seconds in a 40 sccm $O_2$ plasma and consecutively the PMMA resist was dissolved in acetone for 20–30 mins.

## Transmission electron microscopy

Selected area electron diffraction (SAED) patterns were obtained with a 40 μm diameter aperture aperture (defining a selected diameter of ~ 720 nm on the sample) using FEI TitanX TEM operated at 60–80 kV.

Four-dimensional scanning transmission electron microscopy (4D-STEM) was performed on FEI TitanX (80 keV, 0.3 mrad indicated convergence semi-angle) with the Gatan 652 Heating holder for in-situ heating experiments. The 4D-STEM data was analyzed with the py4D-STEM Python package[37]. First, peak detection in py4DSTEM was used to identify the primary Bragg peaks and construct the reciprocal vectors of the host lattice. Next, CDW reciprocal vectors were calculated from symmetry relations to the primary Bragg vectors. Subsequently, we constructed virtual apertures that mask the entirety of the diffraction space except for the CDW ($\alpha$, $\beta$ or IC) satellite peak regions (Supplementary Fig. 6a, b). These CDW virtual apertures were then applied to the 4D-STEM diffraction data to integrate the CDW intensities at each probe position. Note, we only integrated over the CDW satellite peaks around the second and third-order primary Bragg peaks to minimize the diffuse scattering contribution. Further, we also defined a background (bg) virtual aperture to measure the diffuse scattering background for each diffraction pattern (Supplementary Fig. 6a, b). This background was subtracted from the integrated CDW intensities at each probe position for a more accurate calculation of enantiomorphic disproportion. Lastly, for the in-situ heating data, we constructed small virtual apertures that exclude overlap regions between the IC and the $\alpha/\beta$ peaks (Supplementary Fig. 6b).

Differential phase contrast (DPC) STEM images[52] were collected on a Thermo Fisher Scientific Spectra 300 X-CFEG operating at 300 kV with a probe convergence angle of 21.4 mrad. The inner and outer collection angles of the quadrant detector were 15 and 54 mrad respectively. DPC-STEM images were reconstructed from the component images output by each quadrant using the py4DSTEM package[37].

## Raman spectroscopy

Ultra-low frequency (ULF) Raman spectra (Horiba Multiline LabRam Evolution) of $TaS_2$ heterostructures were obtained using a 633 nm laser excitation with the corresponding ULF notch filters at a power of 50–80 μW with 10 s acquisition times and 3 accumulations.

Circularly polarized Raman spectroscopy was performed in backscattering configuration along the ZZ direction. A 100 x objective (N.A. = 0.80) was used for focusing the incident beam onto the sample and for collecting the scattered light. The Raman laser spot size was ~ 1 μm. Circularly polarized light was achieved by using two linear polarizers (LP1 and LP2) and a $\lambda/4$ waveplate (Supplementary Fig. 12).

## Electron transport measurements

Transport measurements were performed using standard lock-in techniques. Briefly, a 1 μA alternating current (17.777 Hz) was applied between the source and drain contacts while sweeping the temperature in the PPMS DynaCool system. Concurrently, the longitudinal ($Vxx$) voltage was measured with the SR830 lock-in amplifier. All phases were ≤ 5, and the resistances were determined from Ohm's law. All data displayed in this work displays the four-probe resistance except for the measurement for S1-R1, which was taken in the 3-probe configuration.

## Density functional theory calculations

Density Functional Theory (DFT) calculations were carried out using the Vienna Ab initio Simulation Package (VASP)[54–57] with projector augmented wave (PAW) pseudopotentials[58,59] including Ta 5$pd$, 6$s$, and S 3$sp$ electrons as valence. The plane-wave energy cutoff was set to 400 eV and the $k$-point grids were Gamma-centered with a $k$-point spacing of 0.3 Å$^{-1}$ for the 1$T$ phase and 0.2 Å$^{-1}$ for the 1$H$ phase, which gave an energy convergence of 1 meV per atom. The convergence criteria for the electronic self-consistent loop was set to $10^{-7}$ eV. Structural optimizations were done using the PBEsol[60] exchange-correlation functional until the residual forces on the ions were less than 0.001 eV Å$^{-1}$.

The energy gain for the interlayer slip observed by DPC-STEM was determined by DFT calculations on mixed phase structures. A shift between 1$H$ and 1$T$ layers of ($2a/3$, $b/3$), where $a$ and $b$ are the 1$H$ lattice parameters, was introduced to eliminate direct S-S overlap. This shift is comparable to the offset between layers in the 6R-$TaS_2$ phase. In a 6-layer 1$H$/1$T$-$\alpha$/1$T$-$\alpha$ stack, the energy gain with interlayer slip was 8 meV Å$^{-2}$.

# Data availability

Raw datasets used to generate figures in the main text are publicly available on Zenodo[61]. Additional data is available from the corresponding author upon request.

# Code availability

Exemplary code used for 4D-STEM data processing is publicly available on Zenodo[62].

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

## Acknowledgements

We thank I. Craig and V. McGraw for helpful discussions, and we acknowledge C. Gammer and S. Zeltmann for developing the 4D-STEM acquisition code for TitanX. This material is based upon work supported by the Air Force Office of Scientific Research under AFOSR Award No. FA9550-20-1-0007. B.H.G. was supported by the University of California Presidential Postdoctoral Fellowship (PPFP) and the Schmidt Science Fellows, in partnership with the Rhodes Trust. Work at the Molecular Foundry, LBNL was supported by the Office of Science, Office of Basic Energy Sciences, the U.S. Department of Energy under Contract no. DE-AC02-05CH11231. Confocal Raman spectroscopy was supported by a Defense University Research Instrumentation Program grant through the Office of Naval Research under award no. N00014-20-1-2599 (D.K.B.). Electron microscopy was, in part, supported by the Platform for the Accelerated Realization, Analysis, and Discovery of Interface Materials (PARADIM) under NSF Cooperative Agreement no. DMR-2039380. This work made use of the Cornell Center for Materials Research (CCMR) Shared Facilities, which are supported through the NSF MRSEC Program (no. DMR- 1719875). The Thermo Fisher Spectra 300 X-CFEG was acquired with support from PARADIM, an NSF MIP (DMR-2039380) and Cornell University. Other instrumentation used in this work was supported by grants from the Canadian Institute for Advanced Research (CIFAR-Azrieli Global Scholar, Award no. GS21-011), the Gordon and Betty Moore Foundation EPiQS Initiative (Award no. 10637), the W.M. Keck Foundation (Award no. 993922), and the 3M Foundation through the 3M Non-Tenured Faculty Award (no. 67507585). K.W. and T.T. acknowledge support from JSPS KAKENHI (Grant Numbers 19H05790, 20H00354, and 21H05233). S.M.R acknowledges support from the SCGSR program, the IIN Ryan Fellowship, and the 3M Northwestern Graduate Research Fellowship. S.H. acknowledges support from the Blavatnik Innovation Fellowship. K.I. acknowledges support from the EPSRC (EP/W028131/1).

## Author contributions

S.H. and D.K.B. conceived the study. S.H. and A.M. fabricated the samples. S.H., B.H.G., M.P.E., and K.C.B. performed the experiments. K.I. and S.M.G. carried out the DFT computations. S.H., S.M.R., and C.O. developed the code for the virtual apertures. T.T. and K.W. provided the hBN crystals. S.H. and D.K.B. wrote the manuscript with input from all co-authors.

## Competing interests

The authors declare no competing interests.
