## [Peer Review File · Nature Communications]

Encoding multistate charge order and chirality in endotaxial heterostructuresREVIEWER COMMENTS

Reviewer #1 (Remarks to the Author):

Authors detail fabrication of verti-lateral endotaxial heterostructure of TaS₂ and remarkably illustrate the step-wise transition in both resistance and achirality using electronic measurements and 4D-STEM and correlate them with existence of endotaxial H-TaS₂ using high resolution DPC-STEM. Overall, the work is impressive, the manuscript is well presented, and contains substantive novelty for publication in Nature Communications. I think the article is well-suited for the journal and is of great interest to the field of 2D quantum materials.

I recommend publication of this manuscript to Nature Communications once following comments are addressed:

1. While the results are impressive and the manuscript is well written, I feel sufficient credit to previous works were not given.

For example, Sung et al. [1] is the first work which demonstrated 2D endotaxy of TaS₂ and endotaxial polytype heterostructure as a new platform to stabilize latent CDW states. In the context of 2D materials, it introduces the concept of endotaxial design. I suggest the authors to include a sentence stating that the recent synthesis of endotaxial TaS₂ offers new mechanisms for control and access of CDW states with a citation that follows on the first paragraph.

[1] S. H. Sung et al., Nat. Commun. 13 413 (2022).

While authors gave credit to ref 37, in terms of demonstration of polarized Raman response to chiral CDWs, Y. Zhao et al.'s [2] recent work really dove deeply and demonstrated chiral-locking across 1T-TaS₂/1T-TaS₂ homointerface. I suggest authors citing this article.

[2] Y. Zhao et al., Nat. Commun. 14, 2223 (2023).

A. Ribak et al., [3] is another work that highlights T/H polytype interfaces and merits credit.

[3] A. Ribak et al., Sci. Adv. 6 aax9480 (2020).

2. Polytype heterostructure presented in this work is thick slab of 1T-TaS₂ separated by H-TaS₂. As the authors mentioned in the first paragraph, stabilization of C-CDW in room temperature occurs for thin or monolayers of 1T-TaS₂. In this context, shouldn't the 1T-TaS₂ presented in this work exhibit NC-CDW phase? For example, in Figure 1c, I would expect to see a mixture of C and NC at least in one of chirality as was shown in Figure S10 of [1], and similarly for resistance vs temperature I expect additional jump in resistance around 200 K. Do you have an explanation for the mechanism of the stabilization of C-CDW in this heterostructure?

[1] S. H. Sung et al., Nat. Commun. 13 413 (2022).

3. In page 9, the authors suggest that the broadening of phase transition is maybe due to strengthened out-of-plane interactions. While this is just a speculation I feel this statement is somewhat contradictory to results regarding heterochirality in Figure 4. In essence, if H-TaS₂ screening is enough to disrupt chiral-locking [2], how could it affect the phase transition?

[2] Y. Zhao et al., Nat. Commun. 14, 2223 (2023).

4. In Figure 2g–j insets, the authors presented a schematic diagram of chirality across the heterostructure. Is stacking sequence of the chirality known or just the net chirality. Some clarification would be nice.

5. I think there is typo in SI. Shouldn't 'SR830 alock-in amplifier' in page 18 of SI be 'SR830 lock-in amplifier'?

Reviewer #2 (Remarks to the Author):

In their manuscript, Husremovic et. al. present an investigation on the optical, phononic, and electrical properties of 1T-TaS₂ crystals subjected to controlled annealing, resulting in the formation of 1H-TaS₂/1T-TaS₂ heterostructures. The authors thoroughly examine the samples using optical microscopy and establish a connection between changes in optical contrast and the tailored properties of the heterostructure. Multiple advanced characterization techniques are employed, all of which confirm the transformation of the original 1T phase into a layered 1T-1H phase with modified CDW (charge density wave) properties. The findings are intriguing and merit publication; however, similar results and concepts have been previously documented, such as in Nature Communications, 13 (2022) 413 and Nature Communications, 14 (2023) 2223. Consequently, due to the lack of sufficient novelty, I am unable to recommend this manuscript for publication in Nature Communications.

Reviewer #3 (Remarks to the Author):

The manuscript by Samra et al. reports the synthesis of highly tunable endotaxial polytype heterostructures of 1T-TaS₂ and 1H-TaS₂ by moderate thermal annealing of nano-thick 1T-TaS₂ flakes. By employing electron microscopy, Raman spectroscopy, and electronic transport measurements, Samra et al. systematically studied the effects of the number of endotaxial metallic 1H-TaS₂ monolayers on the chiral periodic lattice distortion structure and CDW phase transition behavior of 1T-TaS₂ fragments. Furthermore, the authors demonstrate how strain engineering can be used to nucleate the polytype conversions. The research data presented in the manuscript is detailed and logical. Compared to the previous studies, I think this work push the limit of controlling the chirality of 1H/1T-TaS₂ heterostructure and also demonstrates some applications. However, I have some specific comments and questions, which are listed below and should be considered by the author before the paper is accepted.

1) Firstly, I have some concerns about the novelty of the manuscript. The synthesis of heterochiral 1T-TaS₂ vertical heterostructures separated by 1H-TaS₂ layers by annealing of 1T-TaS₂ thin layers, shown in Figure 1, is not a new discovery, as it has been previously reported by Suk Hyun Sung et al. (Nat Commun 13, 413 (2022)). In addition, the method and theory of polarization dependent Raman measurement to detect the chiralities of CDW in 1T-TaS₂, shown in Figure 3, have also been previously reported by H F. Yang et al. (PRL 129, 156401 (2022)).

2) The changes in red optical contrast (ΔOCR) upon annealing are interesting, and this is the first time I have seen such a significant difference in optical contrast between 1T-TaS₂ and 1H-TaS₂ layers. The manuscript has deeply explored the relationship between ΔOCR and the number of 1H-TaS₂ layers, but there is a lack of theoretical explanation for this difference.

3) The manuscript reports that the transition from the 1T phase to the 1H phase was achieved by rapidly cooling to room temperature after annealing at 350 °C for 30 minutes in high vacuum. The annealing temperature here is much lower than the phase transition temperature previously reported by Suk Hyun Sung et al. (720 K, Nat Commun 13, 413 (2022)). The authors should provide some explanations and specific vacuum values.

4) Figure 2e displayed an illustration of the α - β superlattice. How do the authors confirm the slip amount of central Ta atoms in the upper and lower Star of David? This determines the supercell size of the heterochirality vertical superstructure, and it raises the question of whether there are other possible configurations of the α - β superlattice.

5) Figures 3b and 3c compare the Raman optical activity (ROA) of different regions, but this change does not seem obvious. The authors should provide specific values on the vertical axis for readers to see more clearly.

6) Figure 4 explores the effect of 1H-TaS₂ content or the number of 1H-TaS₂-separated 1T-TaS₂ lamellae on the temperature dependence of resistivity. Could the authors provide further discussion on the impact of heterochirality, specifically the relative content of α and β 1T-TaS₂ on the phase transition of CDW?

Table of Contents

Reviewer 1 comments and responses	2
Comment A1:	2
Comment A2:	2
Comment A3:	3
Comment A3i:	3
Comment A3ii:	4
Comment A4:	5
Comment A5:	5
Comment A6:	6
Reviewer 2 comments and responses	7
Comment B1:	7
Reviewer 3 comments and responses	10
Comment C1:	10
Comment C2:	10
Comment C3:	11
Comment C4:	11
Comment C5:	12
Comment C6:	12
Comment C7:	13

Reviewer 1 comments and responses

In manuscript pdf files, changes in response to the Referee 1 comments have been highlighted

Comment A1:

Authors detail fabrication of verti-lateral endotaxial heterostructure of TaS₂ and remarkably illustrate the step-wise transition in both resistance and achirality using electronic measurements and 4D-STEM and correlate them with existence of endotaxial *H*-TaS₂ using high resolution DPC-STEM. Overall, the work is impressive, the manuscript is well presented, and contains substantive novelty for publication in Nature Communications. I think the article is well-suited for the journal and is of great interest to the field of 2D quantum materials. I recommend publication of this manuscript to Nature Communications once following comments are addressed:

We thank the reviewer for their positive comments and insightful comments.

Comment A2:

While the results are impressive and the manuscript is well written, I feel sufficient credit to previous works were not given.

For example, Sung et al. [1] is the first work which demonstrated 2D endotaxy of TaS₂ and endotaxial polytype heterostructure as a new platform to stabilize latent CDW states. In the context of 2D materials, it introduces the concept of endotaxial design. I suggest the authors to include a sentence stating that the recent synthesis of endotaxial TaS₂ offers new mechanisms for control and access of CDW states with a citation that follows on the first paragraph.

[1] S. H. Sung et al., Nat. Commun. 13 413 (2022).

While authors gave credit to ref 37, in terms of demonstration of polarized Raman response to chiral CDWs, Y. Zhao et al.'s [2] recent work really dove deeply and demonstrated chiral-locking across 1T-TaS₂/1T-TaS₂ homointerface. I suggest authors citing this article.

[2] Y. Zhao et al., Nat. Commun. 14, 2223 (2023).

A. Ribak et al., [3] is another work that highlights T/H polytype interfaces and merits credit.

[3] A. Ribak et al., Sci. Adv. 6 aax9480 (2020).

We appreciate the reviewer's comment and acknowledge the importance of giving sufficient credit to previous literature works. To address this comment, **we adjusted the original manuscript** as follows:

- 1. We added the following sentence into the introduction:**

To this end, the recent synthesis of endotaxial TaS₂ offers new mechanisms for accessing modular CDW systems (Sung, 2022).

- 2. We cited A. Ribak et al., Sci. Adv. 6 aax9480 (2020) after the following sentence in the introduction:**

Accordingly, together with flake thickness and doping levels, vertical heterostructuring is a powerful route for engineering CDW transitions.

3. **We cited Y. Zhao et al., Nat. Commun. 14, 2223 (2023)** after introducing circularly polarized Raman of 1T-TaS₂:

The chirality of TaS₂ heterostructures can also be assessed optically, as enantiomorphic disproportion engenders Raman optical activity (ROA): a distinct Raman response to right- and left-circularly polarized light (See SI section 4.2 for details) (Yang 2022; Zhao, 2023).

4. **We introduced the work on 1T-TaS₂ homointerfaces and cited Y. Zhao et al., Nat. Commun. 14, 2223 (2023):**

“...distinguishing them from 1T-TaS₂ flakes/homointerfaces and heavily transformed 1H-TaS₂/1T-TaS₂ heterostructures, which can only manifest a single chirality state (Sung 2022; Zhao, 2023).”

“In pristine 1T-TaS₂ and 1T-TaS₂ homointerfaces, CDW clusters can be perfectly eclipsed because their building blocks—Ta ions—are directly aligned in the out-of-plane direction (Stahl 2020; Zhao 2023).”

Comment A3:

Comment A3i:

Polytype heterostructure presented in this work is thick slab of 1T-TaS₂ separated by H-TaS₂. As the authors mentioned in the first paragraph, stabilization of C-CDW in room temperature occurs for thin or monolayers of 1T-TaS₂. In this context, shouldn't the 1T-TaS₂ presented in this work exhibit NC-CDW phase? For example, in Figure 1c, I would expect to see a mixture of C and NC at least in one of chirality as was shown in Figure S10 of [1], and similarly for resistance vs temperature I expect additional jump in resistance around 200 K.

[1] S. H. Sung et al., Nat. Commun. 13 413 (2022).

We thank the reviewer for the constructive comment. We have now examined the possibility of finding coexisting C-CDW and NC-CDW phases in our moderately transformed heterostructures using selected area electron diffraction (SAED). To this end, we examined heterostructures with different degrees of polytype transformations and drew general conclusions summarized as follows:

We find that all measured samples comprising $\geq 20\%$ of H-TaS₂ manifest an ordered C-CDW phase, characterized by sharp CDW reflections. In contrast, the CDW reflections observed for samples converted to a lesser extent are less well-defined and exhibit peak splitting and angular blurring. The angular blurring suggests the presence of NC-CDW domains, which are rotationally offset from the C-CDW phase by 1.9° . However, our SAED data do not reveal clear hallmarks of the long-range NC-CDW phase, (e.g., 3-fold satellite peaks around the CDW reflections and stronger intensity of higher-order CDW reflections compared to the first order reflections). Thus, we conclude that minimally transformed samples host a disordered C-CDW phase with likely presence of some NC-CDW domains. **We added a writeup of this discussion in the main text and added the representative SAED data for heterostructures of different compositions in SI Figure 4.**

Addition to Main text, Results:

The ensemble room-temperature ordering of CDW domains in TaS₂ heterostructures with different polytype compositions was further probed using SAED. Our findings support that samples comprising $\geq 20\%$ of H-TaS₂ manifest an ordered C-CDW phase, characterized by sharp CDW reflections (SI Figure 4b–d). In contrast, the CDW reflections observed in

samples with < 20% $H\text{-TaS}_2$ appear less well-defined and exhibit peak splitting and angular blurring (SI Figure 4a). Accordingly, we infer that crystals containing less than 20% $H\text{-TaS}_2$ content host a disordered C-CDW phase with likely coexistence of some NC-CDW domains.

Addition to Supporting information:

SI Figure 4. Selected area electron diffraction (SAED) patterns of S3 in regions (a) R2, (b) R3, (c) R4, (d) R5. Scale bars are 5 nm^{-1} .

Regarding the absence of the low-temperature upturn in our samples, we note that current flow would primarily take place through the lower-resistance (metallic) $H\text{-TaS}_2$ layers at these temperatures. So, transport will be dominated by the $H\text{-TaS}_2$. This would preclude the observation of low-temperature upturns in our mixed polytype crystals.

Comment A3ii:

Do you have an explanation for the mechanism of the stabilization of C-CDW in this heterostructure?

In our verti-lateral heterostructures, $1T\text{-TaS}_2$ layers continuously extend across regions with different polytype compositions and corresponding $1T\text{-TaS}_2$ slab thicknesses. We speculate that C-CDW domains in thinner $1T\text{-TaS}_2$ slabs could serve as templates for the nucleation of C-CDW domains in adjacent thicker $1T\text{-TaS}_2$ slabs. However, a comprehensive study on the nano-scale mechanism of C-CDW domain formation in minimally transformed samples lies beyond the scope of this manuscript.

Comment A4:

In page 9, the authors suggest that the broadening of phase transition is maybe due to strengthened out-of-plane interactions. While this is just a speculation I feel this statement is somewhat contradictory to results regarding heterochirality in Figure 4. In essence, if H -TaS₂ screening is enough to disrupt chiral-locking [2], how could it affect the phase transition?

[2] Y. Zhao et al., Nat. Commun. 14, 2223 (2023).

We thank the reviewer for the insightful comment, which is a valid counterpoint to our original hypothesis. Indeed, if we consider that broadening implies defect pinning, an alternative hypothesis for the difference in cooling and warming profiles is that the defect pinning strength is contingent on the CDW phase [*c.f.* 1–4]. Defects may exert stronger pinning effects on the CDW domains in the localized C-CDW state compared to the “melted” IC-CDW phase, leading to less well-defined transitions upon warming. We removed the previous proposal and provided this alternative explanation instead in the main text.

- [1] Ishiguro, Y.; Bogdanov, K.; Kodama, N.; Ogiba, M.; Ohno, T.; Baranov, A.; Takai, K. Layer Number Dependence of Charge Density Wave Phase Transition Between Nearly-Commensurate and Incommensurate Phases in $1T$ -TaS₂. *J. Phys. Chem. C* **2020**, *124*, 27176–27184.
- [2] He, R.; Okamoto, J.; Ye, Z.; Ye, G.; Anderson, H.; Dai, X.; Wu, X.; Hu, J.; Liu, Y.; Lu, W.; Sun, Y.; Pasupathy, A. N.; Tsen, A. W. Distinct Surface and Bulk Charge Density Waves in Ultrathin $1T$ -TaS₂. *Phys. Rev. B* **2016**, *94*, 201108.
- [3] Su, J.-D.; Sandy, A. R.; Mohanty, J.; Shpyrko, O. G.; Sutton, M. Collective Pinning Dynamics of Charge-Density Waves in $1T$ -TaS₂. *Phys. Rev. B* **2012**, *86*, 205105.
- [4] Tsen, A. W.; Hovden, R.; Wang, D.; Kim, Y. D.; Okamoto, J.; Spoth, K. A.; Liu, Y.; Lu, W.; Sun, Y.; Hone, J. C.; Kourkoutis, L. F.; Kim, P.; Pasupathy, A. N. Structure and Control of Charge Density Waves in Two-Dimensional $1T$ -TaS₂. *Proc. Natl. Acad. Sci.* **2015**, *112*, 15054–15059.

Addition to Main text, Results:

“We note that resistance traces upon warming are noticeably broadened relative to the respective cooling sweeps. This may be understood by considering that strength of defect pinning is contingent on the CDW phase [1–4]. Defects may exert stronger pinning effects on the CDW domains in the localized C-CDW state compared to the “melted” IC-CDW phase, leading to less well-defined transitions upon warming. “

Comment A5:

In Figure 2g–j insets, the authors presented a schematic diagram of chirality across the heterostructure. Is stacking sequence of the chirality known or just the net chirality. Some clarification would be nice.

Thank you for pointing out that further clarification is needed. Indeed, we can only determine net chirality from plan-view 4D-STEM. To clarify this point, **we added the following sentence into the caption of Figure 2:**

Addition to Caption of Figure 2:

“We note that only the net chirality can be determined, and the exact stacking sequence of the chirality shown here is one of several possibilities as the precise sequence cannot be obtained from plan-view 4D-STEM.”

Comment A6:

I think there is typo in SI. Shouldn't 'SR830 alock-in amplifier' in page 18 of SI be 'SR830 lock-in amplifier'?

We thank the reviewer for finding this typo, which is now corrected in the Methods writeup.

Main text, Methods section:

“...with the SR830 lock-in amplifier.”

Reviewer 2 comments and responses

In manuscript pdf files, changes in response to Referee 2 comments have been **highlighted**.

Comment B1:

In their manuscript, Husremovic et. al. present an investigation on the optical, phononic, and electrical properties of $1T$ -TaS₂ crystals subjected to controlled annealing, resulting in the formation of $1H$ -TaS₂/ $1T$ -TaS₂ heterostructures. The authors thoroughly examine the samples using optical microscopy and establish a connection between changes in optical contrast and the tailored properties of the heterostructure. Multiple advanced characterization techniques are employed, all of which confirm the transformation of the original $1T$ phase into a layered $1T$ - $1H$ phase with modified CDW (charge density wave) properties. The findings are intriguing and merit publication; however, similar results and concepts have been previously documented, such as in Nature Communications, 13 (2022) 413 and Nature Communications, 14 (2023) 2223. Consequently, due to the lack of sufficient novelty, I am unable to recommend this manuscript for publication in Nature Communications.

We thank the reviewer for their feedback. However, we note that the central results of this work have not been previously documented. Though, we agree that we could do a better job of emphasizing the novelty of our findings more explicitly in the manuscript. Specifically:

1. Novelty of sample structure. In this study, we demonstrate the preparation of novel polytype heterostructures that consist of interdispersed monolayer H -TaS₂ between few-layer $1T$ -TaS₂ lamellae. This is the opposite of the well-studied heterostructures consisting of monolayer $1T$ -TaS₂ sandwiched within thicker $2H$ -TaS₂ blocks. These are key structural differences that lead to considerable changes in optical and electronic properties (distinct from the endotaxial TaS₂ heterostructures studied in any previous work). Notably, moderate heating produces well-defined, coexistent vertical and lateral heterostructures that introduce interesting possibilities for multi-component devices. Such “verti-lateral” polytype heterostructures have not been previously reported on.

2. Realizing reliable, multi-step switching of resistance and chirality in TaS₂ heterostructures. This aforementioned new sample structure presents a distinctive framework that enables the realization of multi-step changes in resistance, structure, and chirality within TaS₂ heterostructures, which have not been reported previously.

Addition to Main text, Introduction:

In this work, we demonstrate an approach converse to preceding literature—interspersing monolayer H -TaS₂ between few-layer $1T$ -TaS₂ lamellae—to isolate CDW transitions and realize the a distinctive framework for deterministic engineering of multistate resistance and chirality changes in TaS₂ above room temperature. Thus, our work provides a versatile and adaptable roadmap for design of reliable, multilevel, multifunctionality switching in phase change materials.

Addition to Main text, Results:

This unprecedented synchronous switching sets the stage for optoelectronic devices combining charge and chirality degrees of freedom.

Addition to Main text, Discussion:

Moreover, the changes in resistance are accompanied by corresponding changes in chirality, resulting in simultaneous modifications of both optical and electrical properties.

3. Expanding the degree of realizable chiral states in TaS₂ heterostructures. As a result of points #1 and #2 above we find that these heterostructures display a wide range of potential overall chiralities, setting them apart from previously studied systems like 1T-TaS₂ flakes/homointerfaces and heavily transformed 1H-TaS₂/1T-TaS₂ heterostructures.

Addition to Main text, Results:

Notably, our verti-lateral heterostructures exhibit a broad spectrum of possible overall chiralities, distinguishing them from 1T-TaS₂ flakes/homointerfaces and heavily transformed 1H-TaS₂/1T-TaS₂ heterostructures, which can only manifest a single enantiomorphic state. Specifically, the former are homochiral, comprising fully of α or β , while the latter have been shown to be achiral, hosting an equal proportion of α and β . Accordingly, our verti-lateral heterostructures may enable chiral opto-electronic memory schemes through their wide array of chiral states that can generate strong optical responses at room temperature.

4. New out-of-plane CDW stacking sequence resulting from interlayer slippage. We show that the interlayer arrangement of CDW clusters in polytype heterostructures results in a staggered arrangement of CDW domains that has not been reported in previous work.

Addition to Main text, Results:

The interlayer arrangement of CDW clusters in polytype heterostructures exhibits notable distinctions compared to both 1T-TaS₂ flakes and homointerfaces. In pristine 1T-TaS₂, CDW clusters can be perfectly eclipsed because their building blocks—Ta ions—are directly aligned in the out-of-plane direction. In contrast, in polytype heterostructures, Ta ions in 1T-TaS₂ slabs across 1H-TaS₂ interfaces must be laterally offset (Figure 1g–j). Thus, CDW clusters in neighboring 1T-TaS₂ slabs must assume a staggered arrangement, resulting in distinct CDW superlattice patterns (2e–f, SI Figure 19).

5. Elucidating the mechanism of polytype formation and mechanical/strain engineering of designer polytype heterostructures. We demonstrate, to our knowledge for the first time, that H-TaS₂ polytype nucleates at macrosocopic flake defects. Further, we develop mechanistic insights into the polytype conversion process with theoretical support and leverage these nanoscale findings to showcase the potential of mechanical and strain engineering for the rational design of vertical and lateral TaS₂ heterostructures. Furthermore, our results demonstrate the feasibility of creating intricate, multi-component device architectures by combining substrate patterning with moderate thermal annealing conditions.

6. Development of a convenient optical contrast method for determining polytype composition. Using atomic resolution differential-phase-contrast scanning transmission electron microscopy (DPC-STEM) imaging, we benchmark the optical contrast of polytype heterostructures, greatly simplifying the characterization and tracking of phase transitions in 1T-TaS₂ and mixed polytype crystals.

Addition to Main text, Discussion:

The polytype composition of these heterostructures can now be conveniently determined using the optical contrast methodology developed in this work, enhancing the accessibility for characterizing and studying complex TaS₂ crystals.

Reviewer 3 comments and responses

In manuscript pdf files, changes in response to Referee 3 comments have been highlighted.

Comment C1:

The manuscript by Samra et al. reports the synthesis of highly tunable endotaxial polytype heterostructures of $1T$ -TaS₂ and $1H$ -TaS₂ by moderate thermal annealing of nano-thick $1T$ -TaS₂ flakes. By employing electron microscopy, Raman spectroscopy, and electronic transport measurements, Samra et al. systematically studied the effects of the number of endotaxial metallic $1H$ -TaS₂ monolayers on the chiral periodic lattice distortion structure and CDW phase transition behavior of $1T$ -TaS₂ fragments. Furthermore, the authors demonstrate how strain engineering can be used to nucleate the polytype conversions. The research data presented in the manuscript is detailed and logical. Compared to the previous studies, I think this work push the limit of controlling the chirality of $1H/1T$ -TaS₂ heterostructure and also demonstrates some applications. However, I have some specific comments and questions, which are listed below and should be considered by the author before the paper is accepted.

We sincerely appreciate the reviewer's positive comments on our work and valuable insights to improve our manuscript.

Comment C2:

Firstly, I have some concerns about the novelty of the manuscript. The synthesis of heterochiral $1T$ -TaS₂ vertical heterostructures separated by $1H$ -TaS₂ layers by annealing of $1T$ -TaS₂ thin layers, shown in Figure 1, is not a new discovery, as it has been previously reported by Suk Hyun Sung et al. (Nat Commun 13, 413 (2022)). In addition, the method and theory of polarization dependent Raman measurement to detect the chiralities of CDW in $1T$ -TaS₂, shown in Figure 3, have also been previously reported by H F. Yang et al. (PRL 129, 156401 (2022)).

We thank the reviewer for their comment. In fact, the control over chirality and multi-state resistivity (the primary discoveries of this work) are completely new. We acknowledge that the novelty of our findings could have been stated more explicitly in parts. As detailed in full for Referee #2 above, the following are the central findings of this work, all of which are new discoveries.

1. **Novel sample structure:** Few layer $1T$ -TaS₂ separated by monolayer H -TaS₂ (the opposite of what has been made previously)
2. **Multi-step resistivity and chirality changes** caused by the specific structure in 1 above.
3. **Demonstration of a range of enantiomorphic states** in $1T$ -TaS₂ heterostructures.
4. **Identification of a new CDW super-supercell.** As this referee mentions in Comment C5, this is new.
5. **Revealing the role of strain in facilitating polytype conversion**
6. **Development of a simple optical contrast method for polytype composition.** As the referee mentions below in Comment C3, this is the first such demonstration in these systems.

In response to this feedback, we have incorporated additional writeups in the main text that explicitly highlight these key discoveries of our study. These are detailed fully in the response to Referee #2.

Comment C3:

The changes in red optical contrast (ΔOCR) upon annealing are interesting, and this is the first time I have seen such a significant difference in optical contrast between $1T\text{-TaS}_2$ and $1H\text{-TaS}_2$ layers. The manuscript has deeply explored the relationship between ΔOCR and the number of $1H\text{-TaS}_2$ layers, but there is a lack of theoretical explanation for this difference.

Thank you for bringing this to our attention. Differences in optical contrast between freestanding $2H\text{-TaS}_2$ and $1T\text{-TaS}_2$ have already been documented [1, 2]. These are related to crystal structure, electronic properties, and dielectric properties. But the discovery that ΔOCR is linearly related to the number of H layers even within mixed $1H/1T$ endotaxial polytypes is indeed new. In response, we provide some explanation for the optical contrast difference between $1T\text{-TaS}_2$ and $H\text{-TaS}_2$ in SI Section 1.2.

- [1] Li, H.; Wu, J.; Huang, X.; Lu, G.; Yang, J.; Lu, X.; Xiong, Q.; Zhang, H. Rapid and Reliable Thickness Identification of Two-Dimensional Nanosheets Using Optical Microscopy. *ACS Nano* **2013**, *7*, 10344–10353.
- [2] Husremović, S.; Groschner, C. K.; Inzani, K.; Craig, I. M.; Bustillo, K. C.; Ercius, P.; Kazmierczak, N. P.; Syndikus, J.; Van Winkle, M.; Aloni, S.; Taniguchi, T.; Watanabe, K.; Griffin, S. M.; Bediako, D. K. Hard Ferromagnetism Down to the Thinnest Limit of Iron-Intercalated Tantalum Disulfide. *J. Am. Chem. Soc.* **2022**, *144*, 12167–12176.

Addition to Supporting information 1.2:

Optical contrast of 2D flakes on SiO_2/Si is strongly influenced by their refractive index and absorption coefficient. These intrinsic properties are shaped by the crystal structure and the resulting electronic structure. Thus, the structurally distinct $1T\text{-TaS}_2$ and $H\text{-TaS}_2$ flakes on identical substrates exhibit significantly different optical contrasts. We employ this property to quantify the number of $H\text{-TaS}_2$ layers formed upon annealing $1T\text{-TaS}_2$ crystals.

Comment C4:

The manuscript reports that the transition from the $1T$ phase to the $1H$ phase was achieved by rapidly cooling to room temperature after annealing at 350 °C for 30 minutes in high vacuum. The annealing temperature here is much lower than the phase transition temperature previously reported by Suk Hyun Sung et al. (720 K , *Nat Commun* **13**, 413 (2022)). The authors should provide some explanations and specific vacuum values.

Indeed, the T -to- H transition initiates at approximately 230 °C [1]. It is important to note that Sung et al. employed a high temperature annealing process, leading to a strong level of polytype transformation (producing monolayers of $1T\text{-TaS}_2$ separated by thick $2H\text{-TaS}_2$). In contrast, our demonstration is that by operating at substantially lower temperatures, moderate levels of polytype transformation can be achieved to create the vertical-lateral heterostructures discussed in this manuscript. **We have included a detailed description of our annealing process in the Methods section.**

- [1] Wang, Z.; Sun, Y.-Y.; Abdelwahab, I.; Cao, L.; Yu, W.; Ju, H.; Zhu, J.; Fu, W.; Chu, L.; Xu, H.; Loh, K. P. Surface-Limited Superconducting Phase Transition on $1T\text{-TaS}_2$. *ACS Nano* **2018**, *12*, 12619–12628.

Main text, Methods:

The annealing of 1T-TaS₂ crystals. The 1T-TaS₂ flakes were annealed in high-vacuum (approximately 10⁻⁷ Torr) by rapidly warming to 120 °C at 60 °C/min with a 5 minute hold. This is followed by heating at 11.5 °C/min to 350 °C and a 30 minute hold at 350 °C, before rapidly cooling to room temperature at 13.5 °C/min.

Comment C5:

Figure 2e displayed an illustration of the α - β superlattice. How do the authors confirm the slip amount of central Ta atoms in the upper and lower Star of David? This determines the supercell size of the heterochirality vertical superstructure, and it raises the question of whether there are other possible configurations of the α - β superlattice.

We appreciate the reviewer's feedback and acknowledge the need for further explanation in our manuscript. Using differential-phase-contrast scanning transmission electron microscopy (DPC-STEM) imaging, we determined that in polytype heterostructures, Ta ions in 1T-TaS₂ slabs are laterally offset across 1H-TaS₂ interfaces (Figure 1g-j). Given that Ta ions are responsible for CDW clusters, it follows that clusters in neighboring 1T-TaS₂ slabs must assume a staggered arrangement, resulting in distinct CDW superlattice patterns (2e-f, SI Figure 19). **We added this information into the main text.**

Addition to Main text, Results:

The interlayer arrangement of CDW clusters in polytype heterostructures exhibits notable distinctions compared to both 1T-TaS₂ flakes and homointerfaces. In pristine 1T-TaS₂, CDW clusters can be perfectly eclipsed because their atomic building blocks—Ta ions—are directly aligned in the out-of-plane direction. Conversely, in polytype heterostructures, Ta ions in 1T-TaS₂ slabs across 1H-TaS₂ interfaces must be laterally offset (Figure 1g-j). Thus, CDW clusters in neighboring 1T-TaS₂ slabs must assume a staggered arrangement, resulting in distinct CDW superlattice patterns (2e-f, SI Figure 19).

Comment C6:

Figures 3b and 3c compare the Raman optical activity (ROA) of different regions, but this change does not seem obvious. The authors should provide specific values on the vertical axis for readers to see more clearly.

We thank you for the comment. **Figure 3 was adjusted accordingly** and the new version is shown below:

Figure 3. Optical detection of chirality in TaS₂ heterostructures. (a) Schematic of a polarization-dependent Raman measurement. (b,c) Raman spectra in the circular contrarotating polarization configurations ($\sigma^+\sigma^-$ and $\sigma^-\sigma^+$) obtained for region R1 (b) and region R3 (c) of a 20-layer sample S4. Lorentzian peak fits and the cumulative fits are displayed. (d) Stacked bar charts displaying the normalized area percentage of $E_g(\text{I})$ and $E_g(\text{II})$ modes measured in the two contrarotating Raman polarization configurations (top: $\sigma^+\sigma^-$, bottom: $\sigma^-\sigma^+$) for R1–R3 of S4. For (b)–(d), x/n denotes the $1H$ -TaS₂ proportion. Data was obtained at room temperature.

Comment C7:

Figure 4 explores the effect of $1H$ -TaS₂ content or the number of $1H$ -TaS₂-separated $1T$ -TaS₂ lamellae on the temperature dependence of resistivity. Could the authors provide further discussion on the impact of heterochirality, specifically the relative content of α and β $1T$ -TaS₂ on the phase transition of CDW?

Thank you for the insightful question. In our verti-lateral heterostructures, decoupled $1T$ -TaS₂ fragments exhibit independent chiral states whose combined contributions determine the overall chirality. This structural chirality does not directly impact the main CDW electronic characteristics, such as the number and size of CDW phase transitions. Instead, these properties are defined by the number of H -TaS₂ layers that electronically isolate the $1T$ -TaS₂ slabs.

REVIEWERS' COMMENTS

Reviewer #1 (Remarks to the Author):

Thank you for your responses to comments and questions raised about the original manuscript. I am pleased to see more complete discussions and descriptions on the experimental results as well as sufficient credit to previous works.

I feel the reviewers' comments were address properly and adequately, and happy to recommend the revised manuscript for publication in Nature Communications.

Reviewer #3 (Remarks to the Author):

The authors have provided acceptable responses to the referee queries and made several useful modifications to the manuscript. I do not have any further mandatory suggested changes.